



**Robust multi-objective optimization under multiple-**
**uncertainties using CM-ROPAR approach: case study of**
**the water resources allocation in the Huaihe River Basin**
Jitao Zhang[1,2,3], Dimitri Solomatine[2,3,4], Zengchuan Dong[1]
[1]College of Hydrology and water resources, Hohai University; Nanjing, 210000, China.
[2]Water Resources Section, Delft University of Technology; Delft, 2628 CD, Netherlands.
[3]IHE Delft Institute for Water Education; Delft, 2628 AX, Netherlands
[4]Water Problems Institute of RAS, Moscow 119333, Russia
*Correspondence to*: Zengchuan Dong (zcdong@hhu.edu.cn)
**Abstract**. Water resources managers need to make decisions in a constantly changing environment
because the data relating to water resources is uncertain and imprecise. The Robust Optimization and
Probabilistic Analysis of Robustness (ROPAR) algorithm is a well-suited tool for dealing with
uncertainty. Still, the failure to consider multiple uncertainties and multi-objective robustness hinder the
application of the ROPAR algorithm to practical problems. This paper proposes a robust optimization
and robustness probabilistic analysis method that considers numerous uncertainties and multi-objective
robustness for robust water resources allocation under uncertainty. The Copula function is introduced for
analyzing the probabilities of different scenarios. The robustness with respect to the two objective
functions is analyzed separately, and the Pareto frontier of robustness is generated. The relationship
between the robustness with respect to the two objective functions is used to evaluate water resources
management strategies. Use of the method is illustrated on a case study of water resources allocation in
the Huaihe River Basin. The results demonstrate that the method opens a possibility for water managers
to make more informed uncertainty-aware decisions.
1. Introduction
Water resources is a natural resource necessary for human survival (Chen et al., 2017) but also a driving
force for social and economic development (Dong and Xu, 2019). Due to the increasing population and
rapid growth of economy, a contradiction between the supply and demand of water resources is becoming
more acute, water quality problems are becoming more prominent, and water resources have gradually
become a bottleneck for socio-economic development (Zhuang et al., 2018). This phenomenon is
particularly evident in rapidly urbanizing and vital agricultural and industrial production watersheds
(Yang et al., 2017). In this category of watersheds, agricultural and industrial production pose a massive
challenge to water resource management (WRM) due to accelerated urbanization and rapid socio-
economic development (Sun et al., 2019). River basin managers must consider water sources in an
integrated manner and decide how to allocate water resources between different water-using sectors and
cities within the basin (Xiong et al., 2020).
Multi-objective optimization (MOO) is an effective method for improving water resources allocation



(WRA) schemes (Lu et al., 2017; Abdulbaki et al., 2017). MOO can provide decision-makers with WRA
options based on their preferences for objectives, which makes it a well-suited decision-making method
for WRM. Ashofteh et al. (2013) constructed a bottom-line-based multi-objective optimization model to
calculate WRA schemes. Habibi Davijani et al. (2016) presented a multi-objective optimal allocation
model of water resources in arid areas based on maximum socioeconomic benefits. However, WRM is
not only a multi-stage and multi-objective problem but also a complex problem involving uncertainties
and risk management (Yu and Lu, 2018). WRM departments often need to face decision challenges under
uncertain conditions (Hassanzadeh et al., 2016; Ren et al., 2019). Climate change and human activities
have led to an increase in uncertainties in rainfall and water demand in the basin and hence to uncertainty
in managing water resource systems (Jin et al., 2020; Ma et al., 2020; Zhu et al., 2019). Uncertain factors
may lead to the risk of water shortage in the basin, so the existing WRA schemes may not be longer
applicable (Keath and Brown, 2009). Therefore, it is important to study WRA under uncertainty.
Previously, several methods were introduced to analyze uncertainty in WRM. Scenario building and
analysis is regarded as an effective method for considering possible future events and analyzing future
uncertainties (Zeng et al., 2019). The fuzzy logic theory is one of the methods to deal with uncertainty,
which describes uncertainty by fuzzifying the decision variables (Nikoo et al., 2013). Two-stage
stochastic programming (TSP) is also an available planning method in optimization under uncertainty
(Li et al., 2020). However, these approaches do not explicitly evaluate the robustness of the WRA options,
although they take into account the uncertainties in WRA.
Robust multi-objective optimization (RMOO) is an effective method for forming robust WRA schemes.
In relation to water, RMOO was actively applied in the field of water supply system (Kapelan et al., 2005;
Kapelan et al., 2006). In the last decade, RMOO has been gradually applied to other areas of WRM.
Yazdi et al. (2015) and Kang and Lansey (2013) applied robust optimization to design wastewater pipes
by considering uncertainties such as climate change, urbanization, and population change. Marchi et al.
(2016) formed stormwater harvesting schemes under variable climate conditions using RMOO. It should
be pointed out however, that in the mentioned approaches the robustness is often "hidden" into the
objective function or constraints and then a common MOO problem is solved that forms a single Pareto
front. This is indeed an effective method to create solution set which in a certain sense is robust. However,
this approach does not explicitly show the relationship between the solution and the uncertainty variables,
which prevents the decision-maker from clearly understanding the impact of uncertainty, which can
influence the decision. To answer this limitation, the procedure "Robust Optimization and Probabilistic
Analysis of Robustness" (ROPAR) has been developed and presented first in (Solomatine, 2012). The
method will generate multiple Pareto fronts, each corresponding to a sample of uncertain variables so
that the statistical characteristics of the uncertainty of the solution can be analyzed. The ROPAR has been
applied in the design of urban stormwater drainage pipes (Solomatine and Marquez-Calvo, 2019) and for
water quality management in water distribution (Marquez Calvo et al., 2019; Quintiliani et al., 2019).
To the best of our knowledge, the presented versions of the ROPAR methodology have the following
limitations:
● ROPAR method has not been applied to the field of WRA.
● ROPAR method only considers the single source of uncertainty: if there are two sources, then
the joint probability of these sources needs to be considered.
● ROPAR method only analyses the variability of one objective under conditions where the
other objective function level is fixed. Although the ROPAR method can provide decision-
makers with a robust solution under certain conditions, it does not take into account the





relationship between the two objective functions.
Based on the above analysis, although the ROPAR method has proven to be suitable for dealing with
uncertainty, it still needs improvement.
In this study, we propose a Copula-Multi-objective Robust Optimization and Probabilistic Analysis of
Robustness (CM-ROPAR) procedure under multiple uncertainties for WRA. The proposed new
procedure of the ROPAR-family considers the joint probability distribution of uncertainties (in this case,
inflows) and enables decision-makers to check the robustness of the two objective functions separately.
The following text is structured as follows.
First, the definition of robustness is presented. Then, the water demand and inflow in the study area was
analyzed. Then, the steps of the CM- ROPAR algorithm and the water resources allocation model are
described in detail. In addition, robustness criteria are chosen to analyze the robustness of the two
objective functions separately. Finally, the applicability of the CM-ROPAR procedure is illustrated on a
case study of the Huaihe River Basin (HRB).

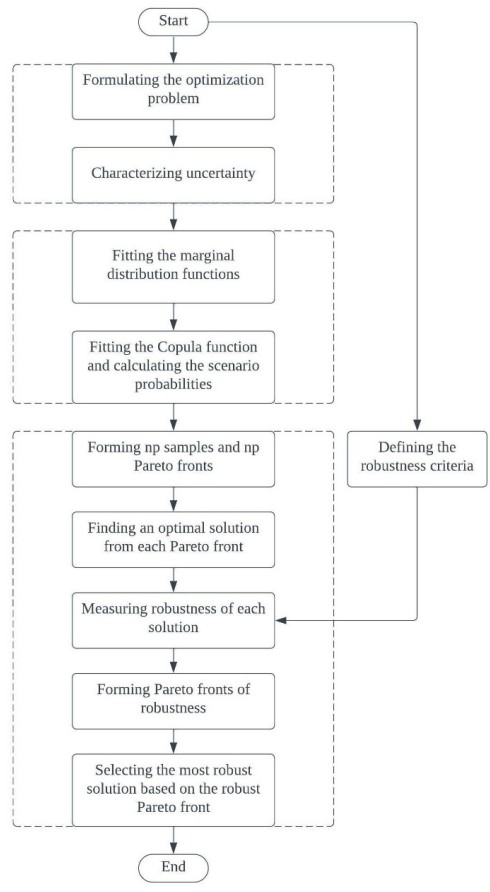


**Figure 1.** Flowchart of CM-ROPAR.
2.    Case Study





The HRB is located in the eastern part of China, and as shown in Figure 2, the middle and upper basin
flows through 15 cities of Henan Province and Anhui Province. It is an important agricultural and
industrial production base in China (Xu et al., 2019). As shown in the Figure 3, the inflow of the HRB
varies significantly between different years and between different regions, and the water demand is
uneven among cities. In addition, due to the discharge of pollutants, the contradiction between supply
and demand of water resources in the middle and upper reaches of the HRB has become increasingly
fierce. Therefore, it is meaningful to study the optimal allocation of water resources and propose a robust
water resources allocation scheme based on the wet-dry encounters in the HRB.

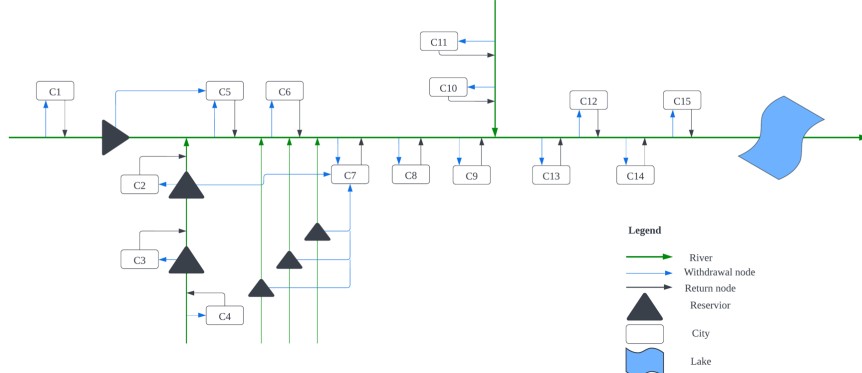

**Figure 2.** Overview of watershed water supply.

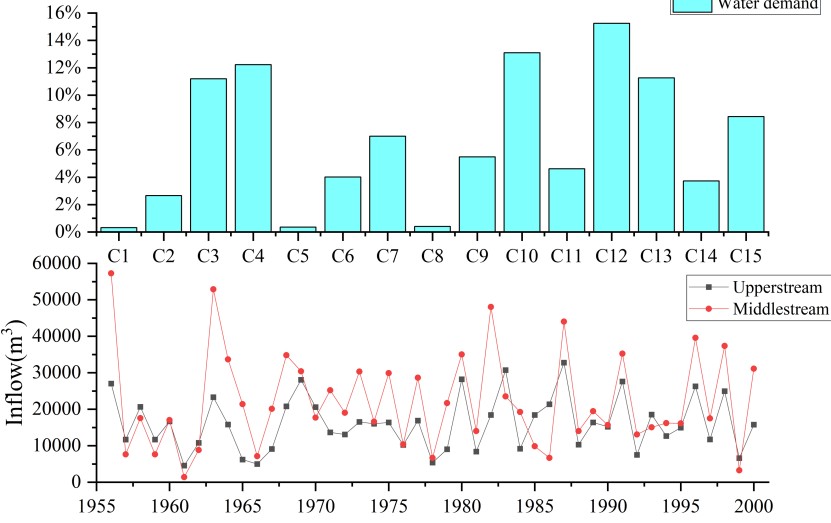

**Figure 3.** Water demand proportion and inflow historical data.
3.   Methodology



### 3.1 Method of Copula Function

Sklar proposed Copula theory in 1959, in which he decomposed an N-dimensional Joint Distribution Function (JDF) into a Copula function and N Marginal Distribution Functions (MDF), which are not required to be the same distribution for N variables and can be used to describe the correlation between arbitrary variables. Nelsen gave a strict definition of Copula function in 1999 (Nelsen et al., 2008). Copula function is the function that connects the JDF with their respective MDF. Copula functions can be expressed as:

$$C_\theta(u_1, u_2 \ldots u_n) = C_\theta[F_1(x_1), F_2(x_2) \ldots F_n(x_n)] \tag{1}$$

where $x_1, x_2 \ldots x_n$ are random vectors, $F_1(x_1), F_2(x_2) \ldots F_n(x_n)$ are MDF of the random vectors, $\theta$ is the parameter of copula function.

Copula functions are mainly classified into Archimedean, elliptic, and quadratic types. Among them, Archimedean Copula functions have been widely applied in the field of hydrology. The most used Archimedean Copula multidimensional joint distribution models are the following:

(1) GH-Copula joint distribution model

$$C_\theta(u_1, u_2 \cdots u_n) = exp\left[-\left(\sum_{i=1}^{n}(-\ln u_i)^\theta\right)^{\frac{1}{\theta}}\right] \; (\theta > 1), \tag{2}$$

(2) Clayton Copula joint distribution model

$$C_\theta(u_1, u_2 \cdots u_n) = \left[1 + \sum_{i=1}^{n}(u_i^{-\theta} - 1)\right]^{-\frac{1}{\theta}} \; (\theta > 1), \tag{3}$$

(3) Frank Copula joint distribution model

$$C_\theta(u_1, u_2 \cdots u_n) = -\frac{1}{\theta}\ln\left[1 + \frac{\prod_{i=1}^{n}(e^{-\theta u_1} - 1)}{(e^{-\theta} - 1)^{n-1}}\right] \; (\theta > 1), \tag{4}$$

The steps of Copula function-based wet-dry encounter analysis are as follows:

1. Fit the MDF. The widely applied probability distribution functions are mainly Pearson type 3 distribution (P-III), T-distribution, Normal distribution, etc.

2. Select the MDF. Fitting different MDF of the runoff, using the AIC criterion for the selection of the fitted MDF.

3. Fitting Copula distribution function.

### 3.2 Method of CM-ROPAR

The CM-ROPAR algorithm consists of the four main parts. The first part is to generate scenarios of drought-wet encounters. The second part is to sample and generate the Pareto front. In this part, the uncertain parameters are sampled firstly. Then a MOO is performed for each sample to generate a Pareto front. The number of Pareto fronts is equal to the number of samples sampled. The third part is a probabilistic analysis of the Pareto front set. The last part is to identify the robust solution. The specific process of optimal water allocation under runoff uncertainty based on CM-ROPAR algorithm is as follows.

**Part 1** (Analyzing the wet-dry encounters)

1.Analyze the inflow wet and dry encounters. If the basin has $k$ inflows, then there are $3^k$ wet-dry scenarios. For example, suppose there is one inflow in the upper and one in the middle reaches of the basin. In that case, there are 9 scenarios: wet-medium, wet-wet, medium-wet, medium-medium, medium-dry, dry-wet, dry-medium, and dry-dry.

2.Choose a scenario from 1 to $3^k$.



**Part 2** (Sampling-Inflow)
3.Based on the recorded annual inflow data $Q$, it is assumed that $Q$ is not a definite value but
$Q = i_{uncertainty} * Q,$ (5)
$i_{uncertainty} \sim N(1,0.0025),$ (6)
where $i_{uncertainty}$ follows a normal distribution with a mean of 1 and a standard deviation of 0.05.
4.For $i = 1,2 \dots, np$ do
5.Sample $u$ (inflow). As mentioned before, the uncertainty variable is obtained from the normal
distribution $N(1,0.0025)$. This represents that a 99.74% probability of the uncertainty variable falling
within the interval $[0.85,1.15]$ and the inflow sample falling within the interval $[0.85 * Q, 1.15 * Q]$.
6.Find the Pareto front $F_r$ by solving the deterministic multi-objective optimization problem for sample
$u_r$.
**Part 3** (Forming the optimal solution set through $np$ Pareto fronts)
7.Select an ideal solution ($IS$) in each Pareto front $F_r$ based on the distance to the origin point, forming
the optimal solution set (set $S$).
**Part 4** (Evaluating the robustness of each solution)
8.Select a solution $s_i$ $(i = 1, \dots, np)$ from the solution set $S$.
9.Cast the inflow case $u_r$ $(r = 1, \dots, np)$ into $s_i$ and calculate $P_r(u_r, s_i)$ and $WD_r(u_r, s_i)$,
respectively, to form 1200 values of $P_r$ and $WD_r$ $(r = 1, \dots, np)$.
10.Select the robustness evaluation criteria, $RC1, RC2, RC3, RC4$.
11.For each $s_i$ $(i = 1,2 \dots, np)$, calculate the $RC1, RC2, RC3, RC4$ and $SRI$ corresponding to $P_r$ and
$WD_r$ respectively. Plot the corresponding graphs and find the Pareto front of each graph.
12.Find the solution with the highest robustness.
End
3.3 Defining the robustness criteria
According to the general definition of robustness, four common Robustness Criteria ($RC$) were used in
this study (Beyer and Sendhoff, 2007). These must be minimized to achieve the maximum robustness of
the solution, so the lower the criteria, the higher the robustness.
For the four $RC$, two MOO are implicitly defined, and optimization can be named Two Layer-Multi-
objective optimization of Robustness Criteria (TL-MOORC). It is worth noting that TL-MOORC differs
from the problem's MOO. A one-layer MOORC is a solution that may not be minimized at all four $RC$
simultaneously. This problem can be solved by aggregating the four $RC$ into one, for example, using a
linear weighted combination. The second layer of MOORC is that for the two objective functions of a
solution, the $RC$ for both objective functions may not be minimized at the same time. Therefore, a trade-
off must be made between the $RC$ for the two objective functions.
The first $RC$ is the expected value of each objective function, denoted as $RC1$. It reflects the fact that
we want to find a solution that is good on average across all uncertainties and can be represented by:
$RC1(s) = \int_{N(s,u)} f(s,u) \, p(u) du,$ (7)
where is the probability density function of the uncertain variable $u$; it is the neighborhood of the
solution $s$.
The second $RC$ is the 'worst case' (or 'minimax' case), denoted as $RC2$. This $RC$ is related to
robustness because we want to find a solution $s$ such that the value of each objective function in the
worst case is the minimum possible. It can be presented as follows:





$$RC2(s) = \min\left(\max_{N(s,u)}\left(f(s,u)\right)\right),\qquad(8)$$
The third $RC$ is the 'standard deviation' of each objective function, denoted as $RC3$. $RC3$ is
related to the robustness of each objective function because we want to find a solution $s$ such that the
value of the objective function would not vary too much due to uncertainty. It can be expressed as follows:
$$RC3(s) = \sqrt{\int_{N(s,u)}\left(f(s,u) - f(u)\right)^2 p(u)du},\qquad(9)$$
The fourth $RC$ is the "probabilistic threshold", denoted as $RC4$. We want to find a solution $s$ that
minimizes the probability that the objective function is higher than the threshold of interest $q$. This
criterion is usually associated with the reliability of the system. It can be expressed as follows:
$$RC4(s) = Pr(f(s,u) > q|s),\qquad(10)$$
In order to evaluate the integrated robustness of the water resources allocation scheme, the weighted sum
of the four Normalized $RC$ ($NRCi$) in this study was used as the integrated robustness criteria. In this
study, we consider that the four $RC$ to be of equal importance, so all four indicators are given a weight
of $\frac{1}{4}$.
$$SRI = \frac{1}{4}NRC1 + \frac{1}{4}NRC2 + \frac{1}{4}NRC3 + \frac{1}{4}NRC4,\qquad(11)$$
(of course, other ways of aggregation can be considered as well.)
3.4  Construction of WRA Model
Objective function
(1)    Social Goals: Water Deficit ($WD$)
$$minf_1(Q) = \sum_{j=1}^{J}\sum_{k=1}^{K}\left(\frac{D_{jk} - \sum_{t=1}^{T}\sum_{i=1}^{I}Q_{ijkt}}{D_{jk}}\right)^2,\qquad(12)$$
Where $D_{jk}$ denotes the water demand of the water consumption department k of the city $j$. $Q_{ijkt}$ is the
water supply quantity of water source $i$ to water consumption department $k$ of the city $j$ in the period
$t$.
(2)    Ecological goals: Pollution ($P$)
$$minf_2(Q) = \sum_{j=1}^{J}\sum_{k=1}^{K}d_{jk}p_{jk}\sum_{i=1}^{I}\sum_{t=1}^{T}Q_{ijkt},\qquad(13)$$
Where $d_{jk}$ denotes the representative pollutant discharge per unit of wastewater of the water department
$k$ of calculation unit $j$ ($ton/m^3$) and $p_{jk}$ represents the sewage discharge coefficient of the water
consumption department of calculation unit. Discharge coefficient of water consumption department $k$
of calculation unit $j$. $Q_{ijkt}$ is the water supply quantity of water source $i$ to water consumption
department of calculation $k$ unit $j$ in the period $t$.
Constraints





(1)     Water demand constraint
$minD_{jk} \leq \sum_{i=1}^{I} \sum_{t=1}^{T} Q_{ijkt} \leq maxD_{jk},$          (14)
(2)     Water supply capacity constraint
$\sum_{k=1}^{K} \sum_{j=1}^{J} \sum_{t=1}^{T} Q_{ijkt} \leq U_i,$          (15)
(3)     Water Resources Constraint
$\sum_{j=1}^{J} \sum_{k=1}^{K} Q_{ijk} \leq WR_i,$          (16)
4.    Results and discussion
4.1   Identification of marginal distribution functions
According to the first part (step 1-2) of the CM-ROPAR process, we need to construct the joint
probability distributions for the upstream and midstream inflow and generate nine inflow scenarios via
the Copula function. Therefore, before constructing the JDF, we need to construct the MDF for the
upstream and midstream inflows respectively. Based on the Kolmogorov-Smirnov (K-S) test results, we
found that the best-fitting distributions for the upstream and midstream were the Weibull and P-III
distributions, respectively.
4.2   Analysis of upstream and midstream dry and wet encounters
The optimal Copula function is selected by comparing the Akaike information criterion (AIC) and the
Bayesian information criterion (BIC), AIC and BIC values in Table 1. It can be concluded that the joint
distribution function of the upper and middle reaches of the HRB is consistent with the joint distribution
of the Clayton Copula function.
**Table 1.** AIC and BIC values for Copula functions.

| | Gaussian | t | Clayton | Gumbel | Frank |
|---|---|---|---|---|---|
| AIC | -20.86 | -18.34 | **-22.69** | -12.47 | -20.03 |
| BIC | -19.06 | -14.73 | **-20.88** | -10.67 | -18.22 |

Substituting the multi-year annual inflow for the upper and middle reaches of the HRB into the Clayton
Copula function, respectively, the following results were obtained.

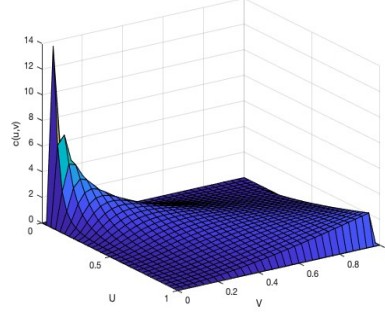
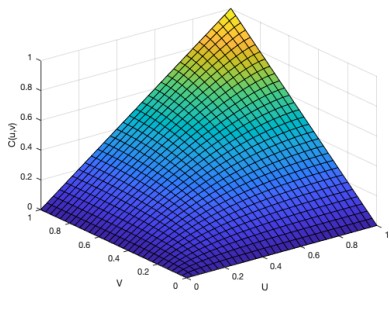

a                                         b

**Figure 4.** Clayton Copula function.
As shown in Figure 4, the joint distribution of the annual incoming water in the upper and middle reaches
of the HRB has symmetry. In addition, the joint distribution of annual water in the upper and middle
reaches has a tail correlation, which indicates a higher probability of simultaneous wetness or drought in



the upper and middle reaches.
**Table 2.** The probabilities of 9 scenarios.

| Wet and Dry encounters/% | | Upstream | | |
|---|---|---|---|---|
| | | Wet | Medium | Dry |
| | Wet | 27.7 | 7.8 | 5.3 |
| Middlestream | Medium | 11.6 | 6.5 | 4.6 |
| | Dry | 4.6 | 7.8 | 24.1 |

As shown in Table 2, the probability of drought-wetness synchronization in the upper and middle reaches
of the HRB is 58.3%, while the probability of asynchrony is 41.7%. The former is 16.6% higher than the
latter, indicating that the upper and middle reaches are less able to complement each other. The joint
distribution has a maximum probability of 27.7% that the upstream and midstream are both wet, and the
risk of water scarcity is minimal under this scenario. The joint distribution has the second-highest
probability of both upstream and midstream being dry at 24.1%, with the highest risk of water scarcity
under this scenario.
**4.3 Considering solutions for the uncertainty of inflow through MROPAR**
In this study the situation when the upper and middle reaches are both wet is considered as a case study.
For deterministic optimization we opted for the NSGA-II algorithm, which is widely used and has good
historical performance (Reed et al., 2013). Inflow uncertainty is modelled by sampling 1200 inflows, as
shown in Figure 5.

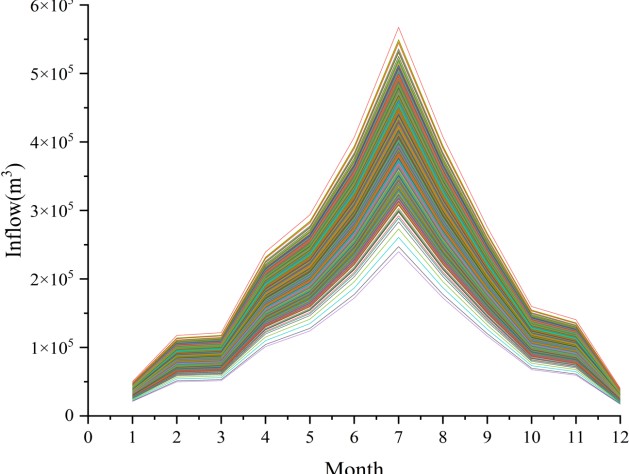

**Figure 5.** Inflow samples.
Figure 6(a) shows that 1200 Pareto fronts calculated for each sampled inflow, through steps 3-6 of CM-
ROPAR. Figure 6(b) shows 1200 ideal solutions $S$, selected based on their distance to the ideal solution
(step 7 of CM-ROPAR).





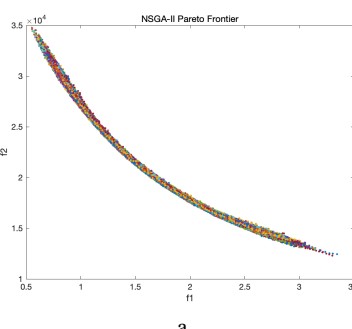
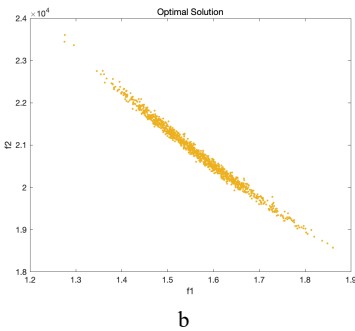

a                                              b

**Figure 6.** a: 1200 Pareto fronts (f1: water deficit; f2: pollution) and b: 1200 ideal solutions (f1: water
deficit; f2: pollution) selected based on their distance to the ideal solution.
**4.4 Assessing robustness of the solutions found by CM-ROPAR**
Four robustness criteria are calculated for each solution $s$ in the solution set $S$. Given the solution $s$
to be evaluated, it is necessary to calculate $WD(s, IF_r)(r = 1, \dots, np)$ and $P(s, IF_r)(r = 1, \dots, np)$ in
order to calculate the four robustness criteria, where $IF_r$ is the $rth$ sample of inflow. $r$ depends on
the number of samples; in this study, 1200 samples were taken, so $np$ is 1200.
As shown in Table 3 and Figure 7, $RC1, RC2, RC3, RC4$ and $SRI$ for $WD$ and $P$ can be calculated
for each solution in $S$, and the solutions corresponding to the smallest value in each $RCi$ and the
solutions corresponding to the smallest value in $SRI$ can be identified, respectively. In addition, we also
feed 1200 samples to the deterministic solution and calculate $RC1, RC2, RC3, RC4$ and $SRI$ for $WD$
and $P$.
**Table 3.** Optimal solution numbers for different robustness criteria.

| | $RC1$ | $RC2$ | $RC3$ | $RC4$ | $SRI$ |
|---|---|---|---|---|---|
| $WD$ | 535 | 361 | 361 | 361 | 361 |
| $P$ | 876 | 876 | 876 | 876 | 876 |
| $IS$ | 629 | 84 | 84 | 915 | 84 |



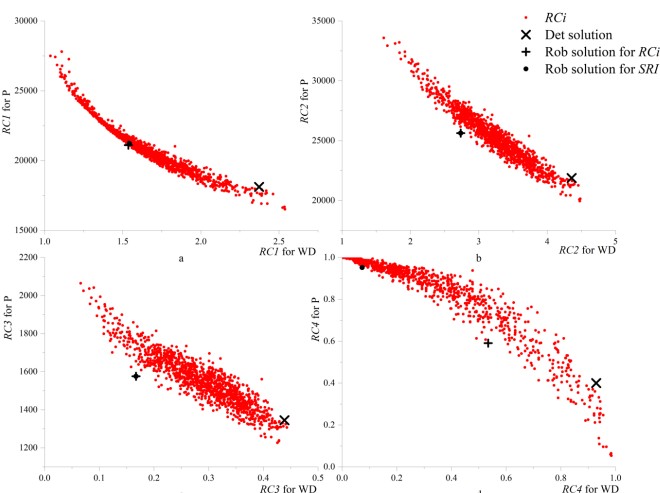


**Figure 7.** Performance of the robustness of solutions (a: $RC1$, b: $RC2$, c: $RC3$, d: $RC4$): robust model
solutions (red dots), deterministic model solution (black ×), solution closest to origin for $RCi$ (black +),
solution closest to origin for $SRI$ (black dot). The horizontal axis represents the performance of the
robustness for $WD$. The vertical axis represents the robustness performance for $P$.

Figure 7 shows the performance of 1200 robust model solutions (red dots) and one deterministic model
solution (black ×), for the four robustness criteria. From Figure 7, four Pareto fronts can also be found,
which indicate the competitive relationship between water deficit and pollution emissions for each
robustness criterion dimension. As shown in Figure 7(a), we can observe an interesting phenomenon that
the left-most extreme solution (red dot) has the smallest robustness index $RC1$ for water deficit, but the
highest robustness index $RC1$ for pollution; the right-most extreme solution (red dot) has the largest
robustness index $RC1$ for water deficit, but the smallest robustness index $RC1$ for pollution. Similarly,
this phenomenon can be also observed for the robustness criteria $RC2$, $RC3$, and $RC4$. More
importantly, as shown in Table 3, the extreme solutions and the solutions closest to the origin point may
differ for different robustness criteria. Specifically, for $RC1$, solution No. 535 is the most robust for
water deficit, and solution No. 876 is the most robust for pollution; for $RC2$, $RC3$, and $RC4$, the most
robust solution for water deficit is solution No. 361, and the most robust solution for pollution is solution
No. 876.

Because there are many non-inferior solutions in the Pareto frontier, the decision-makers must choose
among them. The decision-makers need not only to choose among the non-inferior solutions but also to
evaluate the trade-off between different robustness criteria or to choose the best one by combining the
criteria. This study takes the distance to the origin as the basis for such choice. As shown in Table 3, for
$RC1$, $RC2$, $RC3$, and $RC4$, the closest points to the origin are solution No. 629, solution No. 84, and
solution No. 915, respectively.

**4.5 Comparing solutions found by deterministic and robust approaches**





To see a more general relationship between the 1201 solutions (i.e., 1200 from the robust optimization
solution and 1 from the deterministic optimization solution), the performance of each solution for water
deficit and pollution on each of the four robustness criteria is plotted in Figure 8 and Figure 9.

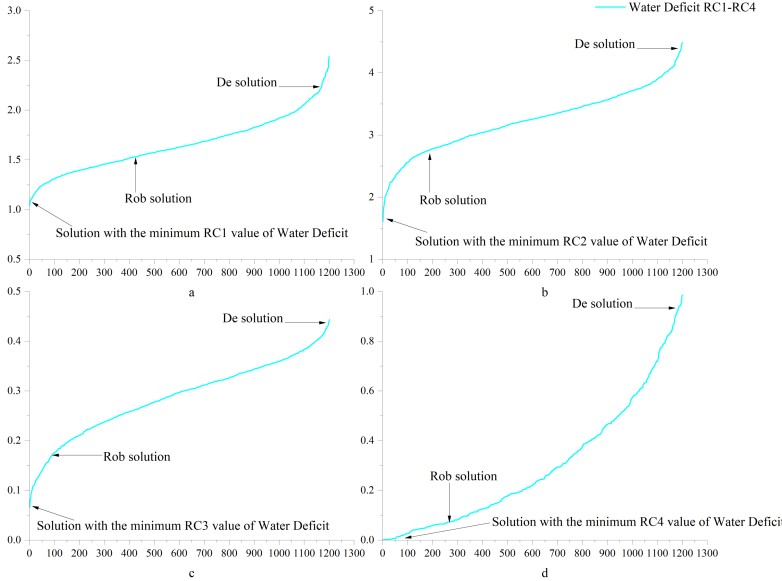


**Figure 8.** Robustness of water deficit (a: $RC1$, b: $RC2$, c: $RC3$, d: $RC4$).

As shown in Figure 8, for water scarcity, the robust solution performed significantly better than the
deterministic solution. Specifically, for the four robustness criteria, the robust solution outperforms
63.1%, 85.6%, 92.7%, and 77.7% of the solutions, respectively, while the deterministic solution
outperforms only approximately 1% of the solutions. To analyze the robust and deterministic solutions
more accurately and intuitively, this study applied the ratio of $RC(Det)/RC(Rob)$ to compare the
robustness of the two solutions. The ratios of $RC(Det)/RC(Rob)$ are 1.53, 1.59, 2.62, and 12.67 in the
four robustness criteria dimensions. This means that, regarding water deficit, the deterministic model
solution may lead to 53%, 59%, 162%, and 1167% more variability in the four robustness criteria
dimensions.

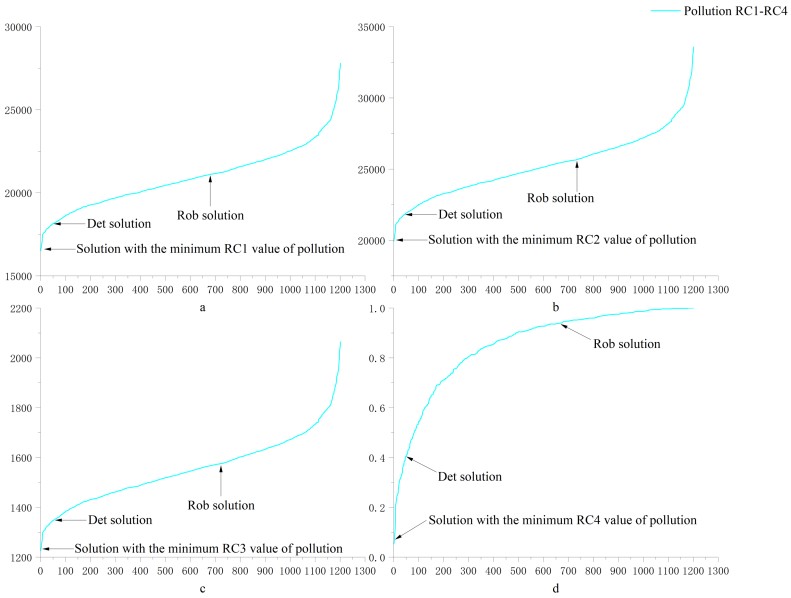

**Figure 9.** Robustness of pollution (a: $RC1$, b: $RC2$, c: $RC3$, d: $RC4$).

However, as shown in Figure 9, the deterministic solution slightly outperforms the robust solution for pollution. Specifically, for the four robustness criteria, the deterministic solution outperforms 96% of the solutions, respectively, while the robust solution outperforms about 40% of the solutions. Similarly, we compare the two solutions by the ratio of $RC(Rob)/RC(Det)$. We find that the $RC(Rob)/RC(Det)$ ratio is about 1.17 for $RC1$ to $RC3$ and 2.37 for $RC4$. This means that, in terms of pollution, the robust solution may lead to 17% more variability for $RC1$ to $RC3$ and 137% more variability for $RC4$.



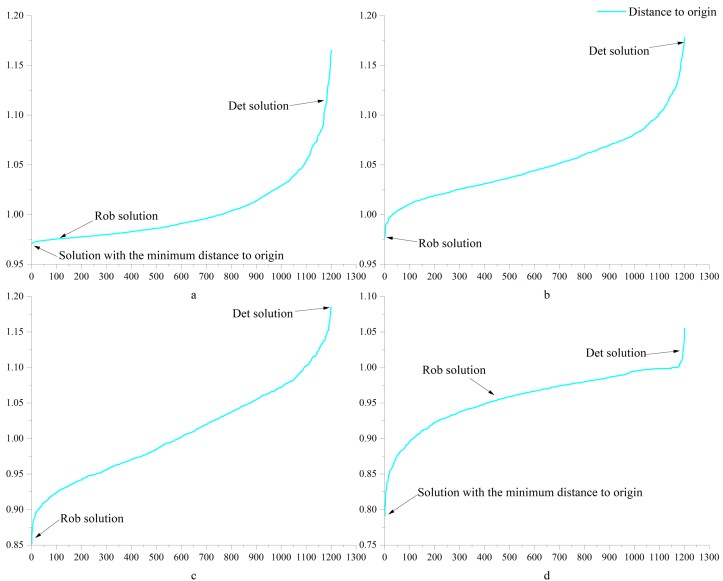

**Figure 10.** Comprehensive robustness for four indicators (a: $RC1$, b: $RC2$, c: $RC3$, d: $RC4$).

In order to analyze the comprehensive performance of each solution, rather than just the robustness of a single objective, this study reflects the comprehensive implementation of each solution in terms of the distance from the solution to the origin. As shown in Figure 10, the comprehensive performance of the robust solution for $RC1$ to $RC4$ is significantly better than that of the deterministic model solution. Specifically, the robust solution outperforms 90.3% and 62.2% of the solutions in $RC1$ and $RC4$, respectively, and outperforms all solutions in $RC2$ and $RC3$, while the deterministic solution performs exceptionally poorly in all four robustness criteria. According to the ratio of $Dis(Rob)/Dis(Det)$, we can find that the robust solution is 16.8%, 19.8%, 39.2%, and 7.3% more robust than the deterministic solution in the four robustness dimensions, respectively.

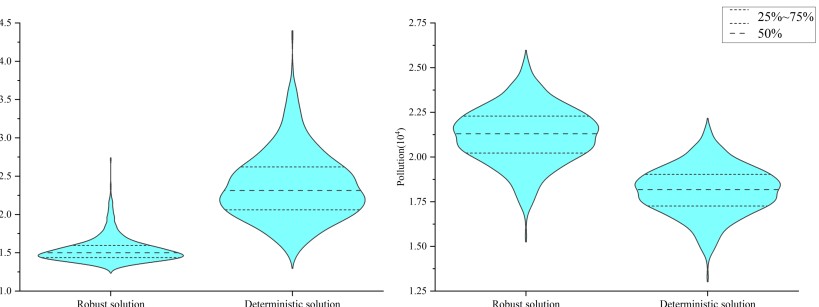

**Figure 11.** The integrated robustness index distribution of the robust and deterministic solution.



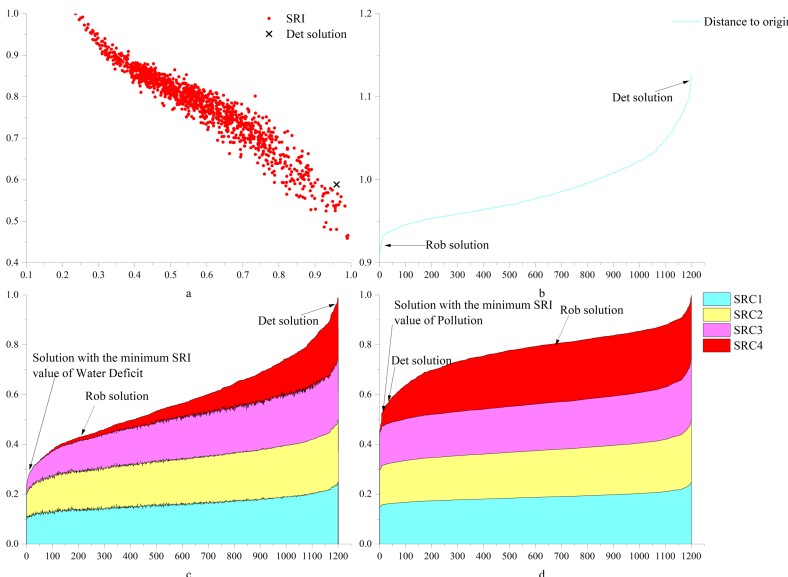


**Figure 12.** Comprehensive robustness criteria performance (a: Performance of comprehensive
robustness criterion, b: Comprehensive robustness of robust solutions and deterministic solution, c and
d: comprehensive robustness criteria for water deficit and pollution).
As shown in Figure 11, for water scarcity, the integrated criteria of the robust solution is clustered at
approximately 0.5 and is significantly more robust than the deterministic solution; for pollution, the
integrated index of the robust solution is significantly higher than that of the deterministic solution, but
the span of the integrated index of the two solutions is similar, so the robustness of the deterministic
solution is slightly better than that of the robust solution.
Similarly, as shown in Figure 12, there is also a Pareto front for the composite robustness criteria. For
water deficit, the robustness of the robust solution is better than the deterministic solution; for pollution,
the robustness of the deterministic solution is better than the robust solution. Specifically, for water deficit,
the robust solution outperforms 85.3% of the solutions while the deterministic solution outperforms only
about 1% of the solutions; for pollution, the deterministic solution outperforms 96% of the solutions
while the robust solution outperforms only 39.6% of the solutions. According to the ratio of
$SRI(Rob)/SRI(Det)$, the deterministic solution is about 130% more uncertain than the robust solution
for water deficit; for pollution, the robust solution is about 37.7% more variable than the deterministic
solution. The distance of each solution to the origin can reflect the comprehensive performance of the
robustness of each solution. For the robustness composite index, the ratio of $Dis(Rob)/Dis(Det)$ is
0.655, which means that the composite robustness of the robust solution is 52.6% higher than the
robustness of the deterministic solution.
For the robustness composite, the robust solution outperforms all the solutions, while the deterministic
model solution outperforms only about 3.2% of the solutions. Comparing the distance to the origin of
the robust solution and the deterministic solution, we can find that the robustness of the robust solution
improves by 27.8% over the deterministic solution.
**4.6 Analysis of specific water resources allocation schemes**



First, as shown in Figure 13, we analyzed the proportion of water supply for each city. We find that the
water supply share for the scheme most robust to water deficit rates is significantly higher than that for
the scheme with the most robust pollutant emissions. This is because an increase in water supply leads
to an increase in pollutant emissions, which in turn leads to a decrease in the robustness of pollutant
emissions. For specific cities, the least robust allocation scenario for water deficit reduces the water
supply in City 3, City 7, City 10, City 12, and City 15 compared to the most robust allocation scenario
for pollutant emissions. Interestingly, these cities have the most water demand in the basin (as shown in
Figure 3). Therefore, basin managers can increase the water supply to these cities if they need to improve
the water deficit robustness of the water resources allocation scheme.

Then we analyze specifically the distribution of water resources between sectors. An interesting
phenomenon can be observed. As shown in Figure 13, although the scenario with the best robustness in
terms of pollutant emissions has a lower water supply than the scenario with the best robustness in terms
of water deficit, the reduction is mainly in the agricultural sector. Water for domestic and industrial
production did not change much. The reason for this may be that agricultural water use causes more
pollution and may create more uncertainty. So how can watershed managers hope that improving the
robustness of pollutant discharge can reduce water supply to the agricultural sector.

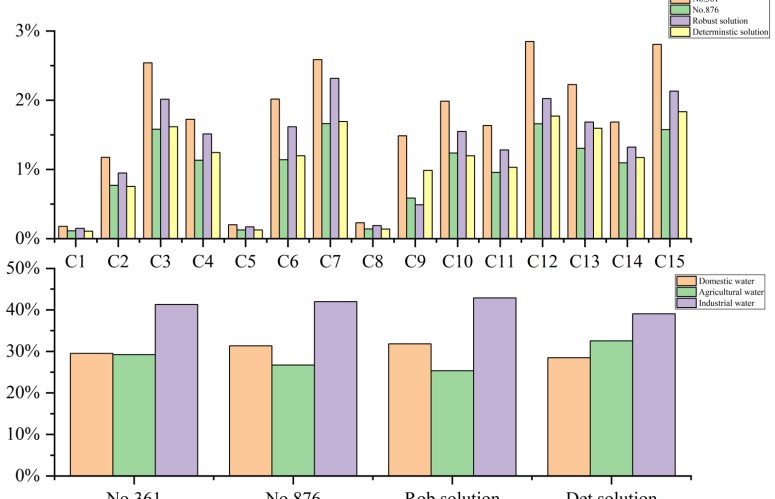

**Figure 13.** Specific water resources allocation schemes.

## 5. Conclusion

In this study, we propose a multi-objective robustness analysis method considering multiple uncertainties
(CM-ROPAR approach) based on the robust optimization method for uncertainty perception (ROPAR
approach). To verify the superiority and practicality of the CM-ROPAR approach, four robustness criteria
are selected, and we compare the robust solution calculated by the method with the optimal solution of
the deterministic model. In the studied case, CM-ROPAR found a more robust solution.

The CM-ROPAR approach permits to exhibit the handling of uncertainty, to be able to analyze how
uncertainty is transmitted to the Pareto frontier, and to perform the corresponding probabilistic analysis.
The novelty of the new method compared to existing ROPAR methods is reflected in two aspects. First,





the ROPAR method only considers uncertainty at a single point. In contrast, the CM-ROPAR method
considers multiple uncertainties through the joint probability distribution of two points, which is closer
to the actual situation and more general. Second, the new way analyzes the robustness of two objective
functions of the solution instead of fixing one objective function to analyze the robustness of the other
objective function. The CM-ROPAR method is more comprehensive and can identify the robustness of
both objective functions, giving decision-makers more information for decision making.
One of the limitations of this study is that the CM-ROPAR approach is applicable to problems with two
uncertainties and two objective functions; however, water allocation allows for more uncertainties and
more objective functions (e.g., the uncertainty of inflow between multiple tributaries). In future research,
we will focus on more complex objective functions and multi-objective optimization problems with at
least three objective functions.
*Author contribution.* JZ and DS conceptualized the study and wrote the paper. ZD provided the data. All
the authors took part in the interpretation of the results and edits of the paper.
*Competing interests.* The authors declare that they have no conflict of interest. Dimitri Solomatine is one
of a member of the editorial board of Hydrology and Earth System Sciences.
*Acknowledgements.* This research has been supported by the projects: Study on the layout of the water
network in Hunan Province (No. XSKJ2021000-05) and Research and application of key technologies
for water resources deployment in the water network system of the Quanmutang Reservoir.
*Data availability.* The code and computed data are available upon request to the corresponding author.

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
