# Peer review of "Robust multi-objective optimization under multiple- uncertainties using CM-ROPAR approach: case study of the water resources allocation in the Huaihe River Basin"

_Hydrology and Earth System Sciences, 2023_

## Referee Comment (RC2)

This paper aims to improve water resources allocation under uncertain inflows. This was achieved by incorporating the Copula function into Robust optimisation and probabilistic analysis of robustness algorithm (ROPAR), so that the uncertainties in multiple inflows can be considered. The paper seems relevant and interesting for the readers of this journal. However, there are several issues to be addressed and clarified before considered publication:

1. Literature review in the introduction needs to be improved to highlight the motivation and the innovation of this study:

(1) In the introduction, some limitation statements are too arbitrary without clearly explaining what previous studies did, and the purpose of doing so. For example, [Line 49-55] The authors stated that several works consider uncertainties in the water resources allocation problem without explicitly evaluating the robustness, and then jumped to the conclusion that they did not explicitly evaluate the robustness. What is scenario building/TSP, and how it is effective for planning?

(2) The authors should highlight and explain "Why does robustness need to be considered?" This is important because there are other Copula-based optimisation studies for water resources allocation but not consider the robustness.

(3) More explanation of the limitations of ROPAR [Line 76-81] is needed to understand the limitations of the current version and the motivation for adding the Copula function into ROPAR. For the limitations, I do not think that ROPAR has not been applied in water resources allocation is a "limitation" of the algorithm itself. And why does the joint probability need to be considered? This should be explained with a context of how the current ROPAR works, and the need for considering joint probability before this paragraph. The authors state that "it does not take into account the relationship between two objective functions" – if ROPAR shows the Pareto front, why does it not consider the relationship between two objective functions?

2. Methods used need to be better explained:

(1) How the Copula function is used in the paper should be explained in more detail.

(2) Why "it follows a normal distribution with a mean of 1 and standard deviation of 0.05" [Line 153]? How are the mean of 1 and standard deviation of 0.05 selected? Is it applied to the case study only or part of the generalisable methodology framework?

(3) How the ideal set of solutions is selected needs to be better explained in detail as this is directly connected to the results in Figure 6.

(4) The statement in Line 174-175 is not correct. Whether robustness criteria is minimised or maximised should depend on the management objectives. For example, if one tries to maximise the water supply reliability, the expected value of the water supply reliability should also be maximised not minimised.

(5) Why use a weighted sum method for normalized RCs?

(6) How is water demand considered in this study? Only Figure 3 shows the demand proportion for each city. Then what is the minimum and maximum water demand in equation (14)?

3. Results also need to be better explained:

(1) What are AIC and BIC [Line 236-237], how are they used in this study and how to read the results of Table 1?

(2) Section 4.3: NSGA-II can also be used in robust optimisation.  How NSGA-II was used in this study, the parameter settings and which input value was used need to be explained. I did not see the optimisation results of using NSGA-II here.

(3) What does Table 3 show? Why is it important to show the number of solutions? This needs to be explained in the text.

(4) Figure 7 shows the robustness values from four different criteria. These are not Pareto fronts. The tradeoffs between the robustness values could be due to the tradeoffs between the original objectives (water deficit and pollution). This is not very surprising. Also, how the solutions are numbered? This might be related to the numbers in Table 3.

(5) Figures 8 and 9 need to be explained on how the x-axis and y-axis means before explaining the results. Also, in Figure 9, the proposed methodology performs worse than the deterministic method if I understand correctly. Then why the proposed method should be used?

(6) What is "comprehensive" performance in Line 329? How are deficit and population considered in this part?

(7) Figure 12 is complicated but important for the readers to understand the results. So it would be important for the authors to clearly explain what the figure is about/how to read the figures before analysing the results.

(8) The beginning of analysis for Figure 13 [Line 366-370] is not very relevant to the figure. The text focuses on the tradeoffs between supply deficit and pollution, which was shown in Figure 6.

(9) Where do different sectors of water (agricultural, domestic and industrial water) come from? How were they considered in the method?

4. The structure of the paper needs to be re-organised:

(1) Figure 1 in the introduction should be part of the methodology if I understand correctly, and needs to be explained in text to help the readers understand the flowchart.

(2) The Methodology should come before the case study, introducing the general framework proposed. And then explain the case study and how the framework is applied to the case study.

There are a lot of details in the methodology that apply only to this case study, for example, Line 143-147.

(3) There is part of the method of the Copula function seems to be a literature review (Section 3.1), which is not suitable for taking up a lot of spaces in the methodology.

(4) Section 3.2: It would be better to compare the CM-ROPAR and the current version of ROPAR with a figure. And the description looks like a mix of algorithm description and the normal description text.

(5) Section 4.1 in the results is part of the method, not the results.

5. Several terms need to be better defined:

(1) How is robustness defined? It may be defined very differently for different areas of study.

(2) [Line 14] multiple uncertainties/ [Line 16] numerous uncertainties: when I first read "multiple uncertainties", I thought this paper deals with multiple sources of uncertainties, such as inflows due to climate variability/climate change, water demand due to population growth, etc. Or it can refer to input uncertainties and model structure uncertainties. And in Line 76-77: the word source is again misleading. But in this paper, "multiple uncertainties" refers to the multivariate uncertainties in multiple inflows in the system.

(3) What is "explicitly" [Line 54] evaluating the robustness and "hidden" [Line 62-63] in the objective function?

(4) What is the joint probability of these sources [Line 77]? How is joint probability defined/calculated? What does the "source" refer to?

(5) What is "wet-dry encounters" [Line 104]? How is it defined?

(6) How is normalized RC [Line 201] defined?

(7) Why is water deficit considered as "Social Goals" [Line 208]?

6. I would suggest the authors polish the writing of the paper, especially focus on the accuracy of certain words and sentences used. This includes but not limited to the following:

(1) Line 13-14: Why ROPAR is a well-suited tool for dealing with uncertainty?

(2) Line 15: What are the differences between the robust optimization proposed and the developed ROPAR?

(3) Line 28: Why mention water quality problems since the paper does not deal with water quality issues?

(4) Line 42: How is multi-stage defined here?

(5) Line 44-46: There is an overlap between this sentence and sentence Line 32-33.

(6) Line 67: "answer" limitation is not correct

(7) Line 97: Here it seems Figure 2 shows the location of the river basin in China from the writing, but Figure 2 shows the conceptual figure of the water allocation scheme.

(8) Line 256: typo in the title: CM-ROPAR, not MROPAR. Please check through the manuscript.

(9) Line 370-371: What is scenario mean here? Is it a certain solution?

(10) There is a lack of explanation for many figures/tables. It would be helpful to add titles, x-axis and y-xis titles and units in the figure, and sentences in text to explain what the figures are about.

---

## Author Comment (AC1)

**Response to Reviewer 1**

Thank you for your thoughtful reading and comments, the authors have benefited from your meaningful suggestions. We have made revisions based on your comments. The parts of the article that were revised we have marked in blue.

**Comment 1:** The novelty proposed by the proposed work is clearly explained in the introduction and consists in the improvement of the ROPAR procedure by introducing multivariate uncertainty analysis by copulas. Nevertheless, I would suggest avoiding bulleted lists to introduce discretional topics such as in lines 75-80.

**Response:** Thank you for your comment. We have revised the part as following: To the best of our knowledge, the presented versions of the ROPAR methodology have the following limitations:(1) ROPAR method has not been applied to the field of WRA; (2) ROPAR method only considers the single source of uncertainty: if there are two sources, then the joint probability of these sources needs to be considered; (3) ROPAR method only analyses the variability of one objective under conditions where the other objective function level is fixed. Although the ROPAR method can provide decision-makers with a robust solution under certain conditions, it does not take into account the relationship between the two objective functions.

**Comment 2:** Please, avoid unnecessary paragraphs, such as line 88.

**Response:** Thank you for your comment. We have revised the part as following: The following text is structured as follows. First, the definition of robustness is presented. Then, the water demand and inflow in the study area was analyzed. Then, the steps of the CM- ROPAR algorithm and the water resources allocation model are described in detail. In addition, robustness criteria are chosen to analyze the robustness of the two objective functions separately. Finally, the applicability of the CM-ROPAR procedure is illustrated on a case study of the Huaihe River Basin (HRB).

**Comment 3:** In general, avoid abusing acronyms, especially the lesser-used ones, as they impede fluent reading.

**Response:** Thank you for your comment. The authors thought that we could not always use the abbreviation for Huaihe River Basin because many foreign readers do not know about Huaihe River Basin, so the authors changed the all HRB to Huaihe River Basin. For the Robustness Criteria(RC), they are used in many places in the paper, and if the full name is used, it may reduce the readability of the paper.

**Comment 4:** What is the function of a figure if it is not described in the text? Where is the flowchart proposed in Figure 1 described? The structure of the paper needs to be supported by the various section references.

**Response:** Thank you for your comment. The original intent of the authors of this part is to introduce the remaining chapters of the article. The flowchart here is that of the CM-ROPAR algorithm, which is clearly inappropriate here. The authors have revised the passage as follows: First, the Chapter 2 presents the methodology of the paper. It mainly includes the method of Copula function, the method of CM-ROPAR algorithm, the definition of robustness and the construction of water resources allocation model.

Then, the Chapter 3 introduces the overview of the study area. Then, the Chapter 4 introduces the application examples of CM-ROPAR algorithm, and this paper is an example of water resources allocation of Huaihe River Basin. Finally, the last Chapter introduces the conclusion of the paper.

**Comment 5:** It would be preferable to maintain a more consonant structure of the manuscript, introducing the case study after the methodology..
**Response:** Thank you for your comment. The authors have changed Chapter 2 to Methodology and Chapter 3 to Case Study.

**Comment 6:** Line 120: "Copula functions are mainly classified into Archimedean, elliptic, and quadratic types." I don't think this statement is true, who states this? There are other widely used copula classes.
**Response:** Thank you for your comment. The authors wanted to express that the basic Copula functions are mainly categorized as Archimedean, elliptic, and quadratic types. However, more than these three are widely used nowadays, for example, the Vine Copula function is also widely used. The authors have revised the passage as follows: The basic copula functions are mainly classified into Archimedean, elliptic, and quadratic types.

**Comment 7:** Section 3.1 describes a general copula analysis without any reference about the proposed work and use of copulas in the analysis.
**Response:** Thank you for your comment. The authors here refer to the 2008 work by Nelson et al. for an introduction to basic Copula function principles. For a reference to the use of copulas, this paper describes how to apply Copula functions to wet and dry encounters in section 3.1.

**Comment 8:** Line 150-153. The introduction of the uncertainty through a normal distribution with mean 1 and sd 0.05 is not clear. Why this distribution and these values?
**Response:** Thank you for your comment. Here is just a case study, other researchers can also set up other distribution forms if they need to use the CM-ROPAR algorithm. Generally speaking, normal distribution is widely used in hydrology fields, and other researchers can set other mean and sd. The authors have revised the passage as follows: As mentioned before, the uncertainty variable is obtained from the normal distribution $N(\mu, \sigma^2)$. Assuming that the uncertainty variable follows $N(1, 0.0025)$, this represents that a 99.74% probability of the uncertainty variable falling within the interval $[0.85, 1.15]$ and the inflow sample falling within the interval $[0.85 * Q, 1.15 * Q]$.

**Comment 9:** The list proposed from lines 143 to 171 is not properly explained. Avoid the technical list without proper explanation, please include the text in paragraphs describing comprehensively the procedure.
**Response:** Thank you for your comment. The authors have improved the presentation by using a more generalized formulation. Especially in the sampling section, we give more generalized cases.

**Comment 10:** The methodology needs to be rewritten and presented in a less confusing way and commented on more comprehensively.

**Response:** Thank you for your comment. The authors have refined this section of the methodology. The first is the structure of the methodology. The authors added a section introducing the principles of drought-wet encounters. Second, the authors rewrote the methodology to be more generalizable.

**Comment 11:** Even if NSGA-II algorithm is a well-known optimizer, please provide more information about the setup of this algorithm, population size, generation, etc.

**Response:** Thank you for your comment. The authors have revised the passage as follows: In this study, the population size is 100, generation is 1000, cross rate is 0.9 and mutate rate is 0.2.

**Comment 12:** The main drawback of the proposed methodology is the lack of flexibility due to the severe limitation such as the number of uncertainty and the object function that can be included in the analysis, both equal to two. Flexibility and easily interpretabilityare crucial characteristics in the decision-making process, for this reason, I would like to know how the authors should overcome this limit and generalise the proposed methodology.

**Response:** Thank you for your comment. The authors believe that your comment is very meaningful. For the CM- ROPAR algorithm, it is not necessary that the uncertainty variable and the objective function are both 2. The objective function can be two or three or more. The number of uncertainty variables we tested 1 uncertainty variable and 2 uncertainty variables. Testing more than two uncertainty variables and more than two objective functions is our next step work.

**Comment 13:** Finally, I suggest mentioning in the conclusion a summary of what comes out from the case study analysis. It could be a benefit for highlighting and quantifying the actual pros of the new proposed methodology.

**Response:** Thank you for your comment. The authors have added a summary of the case studies to Chapter 5 to highlight the superiority of the methodology. The authors have revised the passage as follows: In terms of the study cases in this paper, there is a competitive relationship between the robustness of the two objective functions, which can form a Pareto frontier. For the water deficit rate, the robust solution outperforms the deterministic solution by 53%, 59%, 162%, and 1167% for the four robustness criteria, respectively; for the pollutant emission, the deterministic solution outperforms the robust solution by only 17% for $RC1 - RC3$, and outperforms the robust solution by 137% for $RC4$. For the composite robustness, the robust solution outperforms the deterministic solution by 52.6%.

---

## Author Comment (AC2)

**Response to Reviewer 2**

Thank you for your thoughtful reading and comments, the authors have benefited from your meaningful suggestions. We have made revisions based on your comments. The parts of the article that were revised we have marked in blue.

**Comment 1:** In the introduction, some limitation statements are too arbitrary without clearly explaining what previous studies did, and the purpose of doing so. For example, [Line 49-55] The authors stated that several works consider uncertainties in the water resources allocation problem without explicitly evaluating the robustness, and then jumped to the conclusion that they did not explicitly evaluate the robustness. What is scenario building/TSP, and how it is effective for planning?

**Response:** Thank you for your comment. Past researchers may have treated uncertainty in a fuzzy way or in an internalized way. Although researchers have considered uncertainty in water allocation, the final solution may be an interval rather than a precise one, which may cause confusion for decision makers.

**Comment 2:** The authors should highlight and explain "Why does robustness need to be considered?" This is important because there are other Copula-based optimization studies for water resources allocation but not consider the robustness.

**Response:** Thank you for your comment. It is true that some researchers have calculated wet and dry encounters based on Copula functions when studying water allocation. However, they tend to do such studies under a certain level year, such as under 50% incoming runoff frequency. This paper argues that under climate change conditions, if accurate predictions of future climate cannot be made, robust water allocation schemes need to be proposed to adapt to future climate conditions.

**Comment 3:** More explanation ofthe limitations of ROPAR [Line 76-81]is needed to understand the limitationsof the current version and the motivation foradding the Copula function into ROPAR. For the limitations,I do not think that ROPAR has not been applied in water resources allocation is a "limitation"of the algorithm itself. And why does the joint probability need to be considered? This should be explained with a context of how the current ROPAR works, and the need for considering joint probability before this paragraph. The authors state that "it does not take into account the relationship between two objective functions" –if ROPAR shows the Pareto front, why does it not consider the relationship between two objective functions?

**Response:** Thank you for your comment. The ROPAR family of algorithms is currently used mainly in the design of urban drainage networks, and the authors believe that its current lack of wide application is one of the limitations. The ROPAR algorithm takes into account precipitation at a single area in the design of urban drainage networks, but this is not consistent with the reality, as it is likely that there will be precipitation at multiple locations with inconsistent intensity. In the case of watershed water management, the water inflow to each tributary is different, so the joint distribution needs to be considered.

For the previous ROPAR algorithm, the robust solution is often found after determining the value of a certain objective function and comparing the robust solution with the

deterministic solution. However, in this paper the authors want to find a global robust solution instead of determining a certain objective function value.

**Comment 4:** How the Copula function is used in the paper should be explainedin more detail.

**Response:** Thank you for your comment. The authors thought to describe the principles of the Copula function first in the methodology, and then to present that based on the Copula function we constructed a joint distribution of interval inflows in order to analyze the probability of drought-wet encounters between interval inflows. The authors have revised the passage as follows:

2.2 Method of wet and dry encounters

In a river basin, there may be different drought or wet conditions between different intervals of inflow, so the probability of drought and wet encounters between different intervals of inflow needs to be investigated. According to the analysis in Section 2.1, it is known that Copula function can be used to construct the multivariate joint distribution function. Therefore, this paper adopts Copula function theory to construct the joint distribution and analyze the drought and wet encounter probability.

The steps of Copula function-based wet-dry encounter analysis are as follows:

1. Fit and Select the MDF. The widely applied probability distribution functions are mainly Pearson type 3 distribution (P-III), T-distribution, Normal distribution, etc.

2. Fit and Select Copula distribution function. Fitting different MDF of the runoff, using the AIC and BIC criterion for the selection of the fitted MDF.

3. Calculate the probability of a dry and wet encounters between different interval inflows.

**Comment 5:** Why "it follows a normal distribution with a mean of 1 and standard deviation of 0.05" [Line 153]?How are the mean of 1 and standard deviation of 0.05 selected? Is it applied to the case study only or part of the generalisable methodology framework?

**Response:** Thank you for your comment. Here is just a case study, other researchers can also set up other distribution forms if they need to use the CM-ROPAR algorithm. Generally speaking, normal distribution is widely used in hydrology fields, and other researchers can set other mean and sd. The authors have revised the passage as follows:

As mentioned before, the uncertainty variable is obtained from the normal distribution $N(\mu, \sigma^2)$. Assuming that the uncertainty variable follows $N(1, 0.0025)$,this represents that a 99.74% probability of the uncertainty variable falling within the interval $[0.85, 1.15]$ and the inflow sample falling within the interval $[0.85 * Q, 1.15 * Q]$.

**Comment 6:** How the ideal set of solutions is selected needs to be better explained in detail as this is directly connected to the results in Figure 6.

**Response:** Thank you for your comment. The authors have revised the passage as follows: Each sample produces a Pareto frontier, and there are 100 solutions in each Pareto frontier. After normalizing each objective function, the distance from each solution to the origin is calculated. Based on the distance from each solution to the

origin, the point with the closest distance is selected as the combined optimal solution.

**Comment 7:** The statement in Line 174-175 is not correct. Whether robustness criteria is minimised or maximised should depend on the management objectives. For example, if one tries to maximise the water supply reliability, the expected value of the water supply reliability should also be maximised not minimised.
**Response:** Thank you for your comment. Because the multi-objective models proposed in this paper are minimized, the water supply guarantee rate proposed by the reviewer will also be transformed into the negative of the water supply guarantee rate to be used as the objective function. Therefore, all the robustness metrics proposed in this paper are also minimized.

**Comment 8:** Why use a weighted sum method for normalized RCs?
**Response:** Thank you for your comment. First, the units of the robustness metrics are different for different objective functions, so they need to be normalized. Second, in order to reflect the comprehensive robustness of a solution, so the normalized robustness metrics need to be weighted and summed. In this paper, the four robustness metrics are considered to be equally important, but of course, decision makers can use other methods of calculating weights.

**Comment 9:** How is water demand considered in this study? Only Figure 3 shows the demand proportion for each city. Then what is the minimum and maximum water demand in equation (14)?
**Response:** Thank you for your comment. In this paper, the water demand of each water use unit is calculated by the quota method, and then the water use percentage of each calculation unit is calculated.

**Comment 10:** What are AIC and BIC [Line 236-237], how are they used in this study and how to read the results of Table 1?
**Response:** Thank you for your comment. AIC and BIC stand for Akaike information criterion (AIC) and the Bayesian information criterion (BIC), respectively, and are traditional measures of how well a distribution function is fitted. Smaller values of AIC and BIC mean a better fit.

**Comment 11:** Section 4.3: NSGA-II can also be used in robust optimisation. How NSGA-II was used in this study, the parameter settings and which input value was used need to be explained. I did not see the optimisation results of using NSGA-II here.
**Response:** Thank you for your comment. In this case, 1000 times of multi-objective optimization needs to be done, so NSGA-2 algorithm is used in this paper. As shown in Figure 6(a), this is the 1000 Pareto frontier. In this study, the population size is 100, generation is 1000, cross rate is 0.9 and mutate rate is 0.2.

**Comment 12:** What does Table 3 show? Why is it important to show the number of solutions? This needs to be explained in the text.

**Response:** Thank you for your comment. Table 3 represents the number of the optimal solution corresponding to each robustness metric.

**Comment 13:** Figure 7 shows the robustness values from four different criteria. These are not Pareto fronts. The tradeoffs between the robustness values could be due to the tradeoffs between the original objectives (water deficit and pollution). This is not very surprising. Also, how the solutions are numbered? This might be related to the numbers in Table 3.

**Response:** Thank you for your comment. If we look at the strict definition of a Pareto frontier Figure 7 is indeed not a Pareto frontier, since not all are non-inferior solutions. But if only the outermost points are shown then it can be considered a Pareto frontier. For numbering, the authors have taken the serial number of the program output directly as the number, with no special meaning.

**Comment 14:** Figure 7 shows the robustness values from four different criteria. These are not Pareto fronts. The tradeoffs between the robustness values could be due to the tradeoffs between the original objectives (water deficit and pollution). This is not very surprising. Also, how the solutions are numbered? This might be related to the numbers in Table 3.

**Response:** Thank you for your comment. The horizontal coordinate represents the number of the solution and the vertical coordinate represents the robustness metric value. The robust solution just performs worse than the deterministic solution in objective function 2, but not by much. The intention of this paper is to investigate the relationship between the robustness of the two objective functions, so this paper again proposes the combined robustness in order to facilitate the decision maker to choose the solution.

**Comment 15:** What is "comprehensive" performance in Line 329? How are deficit and population considered in this part?

**Response:** Thank you for your comment. Because the performance of a single objective function on the four robustness dimensions was previously analyzed, the decision maker could not choose a solution. Therefore, this paper proposes the concept of integrated performance.

**Comment 16:** Figure 12 is complicated but important for the readers to understand the results. So it would be important for the authors to clearly explain what the figure is about/how to read the figures before analysing the results.

**Response:** Thank you for your comment. Figure 12 (a) represents the combined robustness metric values of the two objective functions, (b) represents the distance of each solution to the origin to reflect the robustness of each solution, and (c) and (d) represent the weighted values of the two objective functions on the four robustness metrics sorted from smallest to largest.

**Comment 17:** The beginning of analysis for Figure 13 [Line 366-370] is not very

relevant to the figure. The text focuses on the tradeoffs between supply deficit and pollution, which was shown in Figure 6.

**Response:** Thank you for your comment. This paper is analyzing the relationship between water scarcity rates and pollution, but the goal of the CM-ROPAR algorithm proposed by the authors is to develop more robust water allocation schemes for decision makers. Therefore, in this section the percentage of water allocated to each city is analyzed.

**Comment 18:** Where do different sectors of water (agricultural, domestic and industrial water) come from? How were they considered in the method?

**Response:** Thank you for your comment. The decision variable in the multi-objective water allocation model constructed in this paper is the percentage of water allocated to each water sector in each city. Therefore, the water resources allocated to domestic, agriculture, and industry need to be analyzed.

**Comment 19:** Figure 1 in the introduction should be part of the methodology if I understand correctly, and needs to be explained in text to help the readers understand the flowchart.

**Response:** Thank you for your comment. The authors felt that Figure 1 should be placed in the methodology section.

**Comment 20:** Figure 1 in the introduction should be part of the methodology if I understand correctly, and needs to be explained in text to help the readers understand the flowchart.

**Response:** Thank you for your comment. The authors felt that Figure 1 should be placed in the methodology section.

**Comment 21:** The Methodology should come before the case study, introducing the general framework proposed. And then explain the case study and how the framework is applied to the case study.

**Response:** Thank you for your comment. We have changed the second section to a methodology and written the methodology in a more general way. The authors have revised the passage as follows: As mentioned before, the uncertainty variable is obtained from the normal distribution $N(\mu, \sigma^2)$. Assuming that the uncertainty variable follows $N(1, 0.0025)$, this represents that a 99.74% probability of the uncertainty variable falling within the interval $[0.85, 1.15]$ and the inflow sample falling within the interval $[0.85 * Q, 1.15 * Q]$.

**Comment 22:** There is part of the method of the Copula function seems to be a literature review (Section 3.1), which is not suitable for taking up a lot of spaces in the methodology.

**Response:** Thank you for your comment. The authors briefly describe the Copula function.

**Comment 23:** Section 3.2: It would be better to compare the CM-ROPAR and the current version of ROPAR with a figure. And the description looks like a mix of algorithm description and the normal description text.

**Response:** Thank you for your comment. The ROPAR algorithm is to find the solution that is the most robust at this level after the decision maker is given a value for the objective function. The CM- ROPAR algorithm, on the other hand, does not require the decision maker to give the value of an objective function, but rather selects a comprehensive optimal solution from each Pareto frontier and then analyzes the robustness of this solution (robustness under two objective functions; ROPAR analyzes the robustness of only one objective function).

**Comment 24:** Section 4.1 in the results is part of the method, not the results.

**Response:** Thank you for your comment. The marginal distributions need to be preferred, and this part is an explanation of why the Weibull and P-III distributions were selected.

**Comment 25:** How is robustness defined? It may be defined very differently for different areas of study.

**Response:** Thank you for your comment. Robustness, as considered in this paper, is when the objective function does not fluctuate much when the uncertainty variable changes.

**Comment 26:** How is robustness defined? It may be defined very differently for different areas of study.

**Response:** Thank you for your comment. In this paper, robustness is defined as a water allocation scheme that does not lead to drastic changes in water scarcity rates or pollution discharges as the amount of incoming water changes.

**Comment 27:** [Line 14] multiple uncertainties/ [Line 16] numerous uncertainties: when I first read "multiple uncertainties", I thought this paper deals with multiple sources of uncertainties, such as inflows due to climate variability/climate change, water demand due to population growth, etc. Or it can refer to input uncertainties and model structure uncertainties. And in Line 76-77: the word source is again misleading. But in this paper, "multiple uncertainties" refers to the multivariate uncertainties in multiple inflows in the system.

**Response:** Thank you for your comment.

**Comment 28:** What is "explicitly" [Line 54] evaluating the robustness and "hidden" [Line 62-63] in the objective function?

**Response:** Thank you for your comment. Past methods do not provide an obvious indication of the robustness of the solution.

**Comment 29:** What is the joint probability of these sources [Line 77]? How is joint probability defined/calculated? What does the "source" refer to?

**Response:** Thank you for your comment. Here Source represents the source of the two uncertainty variables. The idea expressed here is to study the joint probability distribution of two uncertain variables.

**Comment 30:** What is "wet-dry encounters" [Line 104]? How is it defined?
**Response:** Thank you for your comment. There are often multiple tributaries in a large watershed, and these tributaries are generally not in wet or dry years at the same time. In this case it is clearly unreasonable to assume that the entire watershed is in dry or wet conditions at the same time. Therefore, we need to consider the combination of drought and wetness in different tributaries.

**Comment 31:** How is normalized RC [Line 201] defined?
**Response:** Thank you for your comment. We used a traditional standardized approach. Each solution will have 1000 robustness metric values for 1000 samples. In this paper, each value is normalized by subtracting the minimum value and dividing by the difference between the maximum and minimum values. This normalization ensures that each value is between 0 and 1.

$$NormRC = \frac{RC - minRC}{maxRC - minRC}$$

**Comment 32:** Why is water deficit considered as "Social Goals" [Line 208]?
**Response:** Thank you for your comment. The authors argue that the availability of water for all sectors is tied to the healthy functioning of society, so artificially this is a social goal.

**Comment 33:** Line 13-14: Why ROPAR is a well-suited tool for dealing with uncertainty?
**Response:** Thank you for your comment. We have demonstrated in previous research that ROPAR can be used to deal with uncertainty problems.

**Comment 34:** Line 15: What are the differences between the robust optimization proposed and the developed ROPAR?
**Response:** Thank you for your comment. Past ROPAR methods have considered only one uncertainty variable and have proposed robust solutions after determining a certain objective function value. The CM-ROPAR algorithm proposed in this paper hopes to provide the decision maker with the most comprehensive and robust solution.

**Comment 35:** Line 28: Why mention water quality problems since the paper does not deal with water quality issues?
**Response:** Thank you for your comment. The two objective functions of this paper are to minimize the rate of water shortage and to minimize pollutant discharge, which is clearly related to water quality.

**Comment 36:** Line 42: How is multi-stage defined here.

**Response:** Thank you for your comment. Many scholars have applied multi-stage planning to the field of water allocation. It is generally believed that the first-stage decisions are made before the occurrence of uncertainty events, and after the occurrence of stochastic events, the losses of the whole system are reduced by the second-stage decisions. Specifically in water allocation, the decision maker will often forecast the water demand and then propose a water allocation plan. However, this predetermined decision may not be consistent with the real situation, which requires a second stage to modify the first stage decision to minimize the overall system loss. In the specific case of uncertainty, the interval method is also used to analyze the maximum and minimum values of the uncertain variables, and to calculate the objective function values for the maximum and minimum values of the uncertain variables, respectively.

**Comment 37:** Line 44-46: There is an overlap between this sentence and sentence Line 32-33.
**Response:** Thank you for your comment. The preceding words mainly reflect the impact of human activities and economic development on water resources. The latter reflects the impact of climate change on water resources.

**Comment 38:** Line 67: "answer" limitation is not correct
**Response:** Thank you for your comment. We will use "resolve".

**Comment 39:** Line 97: Here it seems Figure 2 shows the location of the river basin in China from the writing, but Figure 2 shows the conceptual figure of the water allocation scheme.
**Response:** Thank you for your comment. Here the location of each city is similar to the actual situation, and we have added reservoirs and arrows to indicate the water supply. Thus, this map reflects both the location of cities in the watershed and the concept of water allocation.

**Comment 40:** Line 256: typo in the title: CM-ROPAR, not MROPAR. Please check through the manuscript. Line 370-371: What is scenario mean here? Is it a certain solution?
**Response:** Thank you for your comment. We have revised this.

**Comment 41:** Line 370-371: What is scenario mean here? Is it a certain solution?
**Response:** Thank you for your comment. It should be allocation scheme not allocation scenario.

**Comment 42:** There is a lack of explanation for many figures/tables. It would be helpful to add titles, xaxis and y-xis titles and units in the figure, and sentences in text to explain what the figures are about.
**Response:** Thank you for your comment. We will revised the figures.

---

## Author Response (AR2)

**Response to Editors**

The authors thank the editors for being able to give us the opportunity to revise. The authors believe that the suggestions made by the reviewers were meaningful and helpful in improving the readability of our article. We have revised the paper based on the second version. The modifications are marked in red.

The authors have responded to the comments point by point.

Comment 1: What do you mean for "Nelsen gave a strict definition of Copula function in 1999"?
Response: We intended to express that Nelsen described the Copula function in detail and gave some examples. We modify here to (in line 100 and 101): Nelsen discussed the basic properties and some of the main applications of Copula functions in 1999.

Comment 2: In Eq.(1), you do not clarify what are u1, u2, .., un. Please specify this.
Response: We intended to simplify the formula by substituting $u_1, u_2 \ldots u_n$ for $F_1(x_1), F_2(x_2) \ldots F_n(x_n)$, to increase readability. We modify here to (in line 104): $u_1 = F_1(x_1), u_2 = F_2(x_2) \ldots u_n = F_n(x_n)$ are MDF of the random vectors.

Comment 3: Note that, in general, θ is the vector of parameters of the Copula, not the parameter of the copula. In some families, it is a simple parameter, but in general it is not so (for example in vine copulas).
Response: Thanks for the note of caution, here is our negligence. We modify here to (in line 105): $\theta$ is the parameter or the parameter vector of copula function.

Comment 4: The sentence "Among them, Archimedean Copula functions have been widely applied in the field of hydrology" needs the support of references. It's a pity to see that no references about the applications of copulas in hydrology are given in the manuscript.
Response: Thanks for the note of caution, here is our negligence. We quote the book ("Extreme in nature: an approach using copulas" by Salvadori et al. 2007.) in line 108.

Comment 5: The sentence "The most used Archimedean Copula multidimensional joint distribution models are the following" is vague and references are needed here. Did the Authors mean the most used in hydrology?
Response: Thanks for the note of caution, here is our negligence. We intended this sentence as a transitional sentence leading to a specific formula, and also to express that the Archimedean Copula function has been widely used in hydrology.

Comment 6: The sentence "The most used Archimedean Copula multidimensional joint distribution models are the following" is vague and references are needed here. Did the Authors mean the most used in hydrology?
Response: Thanks for the note of caution, here is our negligence. We intended this

sentence as a transitional sentence leading to a specific formula, and also to express that the Archimedean Copula function has been widely used in hydrology.

Comment 7&8: - Check the range of the parameter theta in Eq.(3).
- In Eq. (4) there are mistakes including the range of the parameter theta. Please revise it.

Response: Thanks for the note of caution, here is our negligence. We modify here to(in line 113 and 115):

Clayton Copula joint distribution model

$$C_\theta(u_1, u_2 \cdots u_n) = \left[1 + \sum_{i=1}^n (u_i^{-\theta} - 1)\right]^{-\frac{1}{\theta}} \ (\theta \in [-1,\infty)\backslash\{0\}),$$

Frank Copula joint distribution model

$$C_\theta(u_1, u_2 \cdots u_n) = -\frac{1}{\theta} ln \left[1 + \frac{\prod_{i=1}^n (e^{-\theta u_1}-1)}{(e^{-\theta}-1)^{n-1}}\right] \ (\theta \in R\backslash\{0\}),$$

Comment 9: Lines 119-124: Please revise this part. When the Authors say: "Fit and Select MDF", please specify how they do the fit (method of estimation of parameters). In addition, it is not clear if they make goodness of fit tests for marginals and for the copulas. Please clarify these issues.

Response: Thanks for the note of caution, here is our negligence. We used the k-s test and RMSE values as a test of the goodness of fit of the marginal distribution. We modify here to (In line 123-125.): MDF can be fitted by Maximum Likelihood Estimation method (MLE method) and the goodness-of-fit test can be performed by the Kolmogorov-Smirnov test (K-S test) and the Root Mean Square Error value (RMSE value).

In addition to this, we added the specific goodness-of-fit test table box in line 251.

**Table 1.** MDF goodness-of-fit test results.

| | Distribution type | Upper stream inflow | Middle stream inflow |
|---|---|---|---|
| | Normal | 0.3341 | 0.8637 |
| | Log-normal | 0.5175 | 0.5703 |
| p-value | P-III | **0.7674** | 0.7599 |
| | Weibull | 0.5758 | **0.9658** |
| | Rayleigh | 0.6123 | 0.2173 |
| | Normal | 0.13721 | 0.086144 |
| | Lognormal | 0.11821 | 0.1152 |
| D-value | P-III | **0.0958** | 0.0965 |
| | Weibull | 0.1129 | **0.0708** |
| | Rayleigh | 0.1096 | 0.1533 |
| | Normal | 0.0345 | 0.0522 |
| | Lognormal | 0.1391 | 0.1152 |
| RMSE | P-III | **0.0306** | 0.0358 |
| | Weibull | 0.0929 | **0.0306** |
| | Rayleigh | 0.0529 | 0.1736 |

Because we have added a table, the subsequent table numbering has also changed.